# Identifying novel genetic variants in epidermolysis Bullosa among Middle Eastern Arab Families: Insights from whole exome sequencing and computational analysis

Nancy Shehata[1]*, Babajan Banaganapalli[2,3], Hadiah Bassam Al Mahdi[4], Shmoukh Alghuraibi[4], Mahmoud Younis[5], Sultana Abdulghani[1], Noor Ahmad Shaik[2,3]*, Zuhier Awan[6], Fahad Hakami[5,7]

1 Department of Dermatology, King Abdullah Medical Complex, Jeddah, Saudi Arabia, 2 Department of Genetic Medicine, Faculty of Medicine, King Abdulaziz University, Jeddah, Saudi Arabia, 3 Princess Al-Jawhara Al-Brahim Centre of Excellence in Research of Hereditary Disorders, King Abdulaziz University, Jeddah, Saudi Arabia, 4 Department of Research and Development, Al Borg Diagnostics, Jeddah, Saudi Arabia, 5 Department of Molecular Genetics, Al Borg Diagnostics, Jeddah, Saudi Arabia, 6 Department of Clinical Biochemistry, Faculty of Medicine, King Abdulaziz University, Jeddah, Saudi Arabia, 7 King Abdullah International Research Centre, King Abdulaziz Medical City, Jeddah, Saudi Arabia

* Nancyshehata@gmail.com (NS); nshaik@kau.edu.sa (NAS)

## Abstract

### Background

Epidermolysis Bullosa (EB) is a rare genetic disorder that results in fragile skin and blistering and may lead to mucous membrane involvement. The disease manifests in several subtypes, among which the most serious conditions are dystrophic and junctional EB. This study intends to highlight the recurrent and novel genetic abnormalities that cause EB in the Western region of Saudi Arabia.

### Methods

Twelve Middle Eastern Arab families affected by Epidermolysis Bullosa (EB) were recruited from dermatology clinic from King Abdullah Medical Complex in Jeddah. Detailed clinical phenotyping was conducted for each patient to document EB-associated symptoms and to accurately determine the disease subtypes. Whole Exome Sequencing (WES) was performed to identify genetic variants associated with EB, and the resulting variants were classified by the guidelines of the American College of Medical Genetics and Genomics (ACMG). Additionally, multiple bioinformatics tools were employed to evaluate the pathogenicity of the detected variants. Variant segregation with disease phenotype was confirmed within the families using Sanger sequencing.

**Data availability statement:** All relevant data are within the paper and its Supporting information files.

**Funding:** The author(s) received no specific funding for this work.

**Competing interests:** The authors have declared that no competing interests exist.

## Results

We identified 11 genetic variants, including three novel variants, in the *COL7A1* (NM_000094.4), *COL17A1* (NM_000494.4), and *LAMB3* (NM_000228.3) genes across 12 EB families. The *COL7A1* variants included frameshift variants (c.5924_5927del and c.6268_6269del), nonsense variants (c.1633C>T, c.1837C>T, c.2005C>T, and c.5888G>A), missense variants (c.4448G>A and c.8245G>A), and splice-site variants (c.6751-1G>A and c.8305-1G>A). Additionally, a splice-site variant was identified in *COL17A1* (NM_000494.4; c.1394G>A) and another in *LAMB3* (NM_000228.3; c.1977-1G>A). Bioinformatics analysis predicted these variants to be likely pathogenic because they disrupt collagen VII, XVII, and laminin 332, proteins essential for skin stability. Frameshift and nonsense variants introduce premature stop codons, leading to truncated or degraded transcripts. Splice-site variants likely cause aberrant splicing, disrupting the reading frame and impairing protein function.

## Conclusion

WES is an effective first-line diagnostic tool for identifying EB-associated variants. This study reveals locus and allelic heterogeneity in EB cases from Saudi Arabia. The findings underscore the importance of early genetic screening for improving genetic counseling in high-consanguinity populations and emphasize the need for large-scale genetic studies in the country.

## 1. Introduction

Epidermolysis Bullosa (EB) is a rare inherited dermatosis characterized by skin and mucous membrane blisters and fragility. Symptoms include fragile skin, blisters, sores, scarring, thickened skin, deformities such as mitten deformity, chronic wounds, pain, dental abnormalities, eye problems, and internal complications in severe cases. Epidemiological data regarding the incidence and prevalence of EB vary globally, with reported prevalence rates ranging from 11 to 20 per million individuals in different countries [1–3]. It is highly diverse in presentation, and according to the layer of skin involved, it is classified into four main types: epidermolysis bullosa simplex (EBS, OMIM:131800) (intraepidermal), junctional epidermolysis bullosa (JEB, OMIM: 226700) (within the lamina lucida of the basement membrane), dystrophic epidermolysis bullosa (DEB, OMIM:131750) (below the basement membrane), and Kindler epidermolysis bullosa (KEB, OMIM: 173650) (mixed skin cleavage pattern) [4]. Among these EB forms, the EBS form is more prevalent (92%) worldwide compared to the other minor forms, such as DEB (5%) and JEB (1%) [5]. The numbers might differ in Saudi Arabia, with a retrospective study conducted in the central region reporting 42.9% of EB cases as DEB, 21.4% as JEB, and 35.7% as EBS [6]. The figures were similar in the Eastern region, where the prevalence of DEB was 62.5% and 25% for EBS [7]. A multicenter retrospective review from 2003-2020 reported EBS in 61.2% of cases, DEB in 19.7%, JEB in 16.4%, and KEB in 2.6% [8].

EB is caused by variants in genes that encode when the proteins essential for the skin's structure and integrity are affected. These proteins maintain the adhesion between the epidermis and the dermis. The layer of the skin where blisters form depends on the location of the altered protein [4]. In EBS, blisters form within the basal keratinocytes. In JEB, they arise in the lamina lucida. In DEB, blisters form in the sublamina densa. The cleavage in KEB occurs below the lamina densa and below the lamina lucida in the basal keratinocytes [4,9,10]. Over 29 genes have been associated with EB [11]. Gene-EB phenotype associations have been documented in several studies. For example, 60-70% of the EBS-causing variants affect the *KRT5* (OMIM:148040) and *KRT14* (OMIM:148066) genes, which encode keratins 5 and 14, respectively [4]. Variants in the *LAMA3* (OMIM:600805), *LAMB3* (OMIM:150310), and *LAMC2* (OMIM:150292) genes that result in reducing or eliminating laminin 332, a protein that bridges hemidesmosomes and anchoring fibrils, lead to typical JEB phenotypes [4,12]. Additionally, a variant in *COL17A1* (OMIM: 113811) introduces a premature stop codon in type XVII collagen, leading to JEB [13]. Variants in *COL7A1*(OMIM:120120) that encode collagen VII, a major constituent of anchoring fibrils, produce altered or missing anchoring fibrils and lead to the dystrophic forms of EB [4]. Kindler's EB is caused by variants in the *FERMT1* gene (OMIM: 607900), which encodes kindlin-1, a protein linked to integrins and focal adhesions.

The epidemiology of EB in Middle Eastern populations remains underexplored [14]. Epidermolysis bullosa (EB) is a rare genetic skin disorder in Saudi Arabia. A retrospective study in the Eastern Province identified 16 cases among 49,902 dermatology patients over seven years, with 87.5% of these cases having a history of parental consanguinity (PMID: 8407073) [14]. Another study at King Abdulaziz Medical City in Riyadh analyzed 28 Saudi patients with EB, finding that dystrophic EB (DEB) was the most prevalent subtype, accounting for 42.9% of cases, and identified 14 novel mutations (PMID: 35222512) [15]. These findings highlight the need for comprehensive epidemiological studies to better understand EB's prevalence and genetic diversity in the Saudi population. A few studies have reported a predominance of recessive EB subtypes [16], likely due to the high rate of consanguinity in the region [17]. Variants identified in Saudi EB cases differ from those reported in other geographic regions [18]. However, molecular epidemiology data from the Middle East remains limited [19]. This study aims to advance research in Saudi Arabia by identifying common inherited variants to support improved genetic counseling and future family planning.

Despite increasing global awareness of genetic variants and personalized medicine, only a few studies regarding inherited skin disorders like EB have been conducted on Arab populations living in Saudi Arabia. To address this gap, our study aims to identify genetic causes and characterize EB subtypes in Arab patients using advanced DNA sequencing techniques. We seek to identify disease-causing variants through molecular screening and apply computational analysis to evaluate their impact on protein structure and function.

## 2. Methodology

### 2.1. Family recruitment and clinical investigation

This study was conducted from July 1 to September 13, 2024, following ethical approval from the Biomedical Ethics and Research Committee (Reference number: N06/24). Patients with a confirmed clinical diagnosis of EB were recruited from the Dermatology Clinic of King Abdullah Medical Complex in Jeddah, Saudi Arabia, and referred to Al Borg Diagnostics for genetic testing. Inclusion required written informed consent from patients and their families for both genetic testing and the publication of research findings. Exclusion criteria included concurrent medical conditions that could interfere with participation, enrollment in other clinical trials during the study period, and pregnancy or lactation. Genetic analysis was performed to identify causative variants and correlate them with clinical phenotypes, aiming to enhance the understanding of EB in the Saudi population. Clinically, EB was classified into three main subtypes: epidermolysis bullosa simplex (EBS), presenting with localized blistering on the hands and feet, typically without scarring and with rare mucosal involvement; junctional EB (JEB), marked by severe, life-threatening blistering in infancy and frequent mucosal involvement; and

dystrophic EB (DEB), characterized by scarring blisters, nail dystrophy, milia, and pseudosyndactyly, with severe forms linked to a heightened risk of aggressive squamous cell carcinoma.

## 2.2. Genetic analysis

**2.2.1. DNA isolation.** Peripheral blood samples were obtained from each participant in the study. DNA extraction was performed following the protocol provided with the QIAamp DNA Mini Kit (Qiagen, Alameda, CA, USA). The DNA extracted was then assessed for quality, quantity, and integrity using a Denovix DS-11 spectrophotometer and 1% agarose gel electrophoresis.

**2.2.2. Whole Exome Sequencing (WES).** Genomic DNA (2ug) from the index cases was used for WES analysis, targeting the exonic regions of over 20,000 genes along with the corresponding exon-intron boundaries (+/-15 nucleotides). Enrichment was performed using the Agilent SureSelect V6 kit (Agilent Technologies, USA). Sequencing was performed on an Illumina NextSeq sequencer (Illumina, USA) with an average coverage depth of 100-130X. The raw sequencing data were aligned to the GRCh37/hg19 genome assembly, and variant calling and annotation were conducted using the CLC Genomics workbench and Qiagen bioinformatics tool. Variants with poor quality metrics, such as low sequencing depth (e.g., <30x coverage), high base call error rate (e.g., >1%), low Phred scores (e.g., <30), high duplicate read rate (e.g., >20%), and those with a high frequency in the general population (>1.0%, except for known common pathogenic variants) were excluded. The remaining identified interesting variants were evaluated and classified according to the ACMG guidelines [20]. The workflow details are shown in Fig 1.

**2.2.3. Variant validation and familial segregation.** Sanger sequencing was performed to confirm the candidate-identified variant in the index and to assess its segregation with disease in other family members. Polymerase Chain Reaction (PCR) was conducted using DreamTaq PCR Master Mix (catalog # K9021) on a VeritiPro 96-well thermal cycler from Applied Biosystems (Life Technologies, CA). Primer sequences for the PCR reaction were designed with Primer3Plus (https://www.bioinformatics.nl/cgi-bin/primer3plus/primer3plus.cgi) (S1 Table). Following PCR amplification, Sanger sequencing was conducted in SeqStudio™ Genetic Analyzer (Applied Biosystems, USA). Sequence alignment and variant identification were carried out using SnapGene software, version 6.0.2(https://www.snapgene.com/updates/snapgene-6-0-2-release-notes)

## 2.3. Computational analysis

**2.3.1. Variant predictions.** We further explored the potential impact of the identified genetic variants using several bioinformatics tools. For missense, insertion, and deletion (indels), stop-gain (nonsense), and splice site variants, we used the Combined Annotation Dependent Depletion (CADD) scoring system. Variants with scores of 20 or higher were considered potentially pathogenic [21]. We applied FATHMM to predict functional consequences for missense variants, considering scores > 0.5 indicative of a significant impact [22]. For splice site variants, Splice AI was used to evaluate potential disruptions in normal splicing, with scores >0.5 suggesting likely splicing alterations [23].

**2.3.2. RNA secondary structure analysis.** To examine how splice site variants might affect RNA secondary structure, we used RNAfold from the ViennaRNA package [24]. We looked at regions extending 1,000 base pairs upstream and downstream of each splice site variant. Using RNAfold, we predicted changes in RNA folding, focusing on alterations in minimum free energy (MFE) and structural motifs. Variants that resulted in more than a 10% change in MFE were considered to have a significant impact on RNA structure, potentially influencing RNA processing or stability.

**2.3.3. Protein structure analysis.** We explored the structural impact of the variants on the proteins using the AlphaFold2 server [25]. We created protein models for both the wild-type and mutant forms. By comparing these models, we looked for changes in protein folding, stability, or the configuration of active sites. We quantified these alterations using root mean square deviation (RMSD). Changes exceeding 1.5 Å were considered significant, suggesting potential effects on protein function.

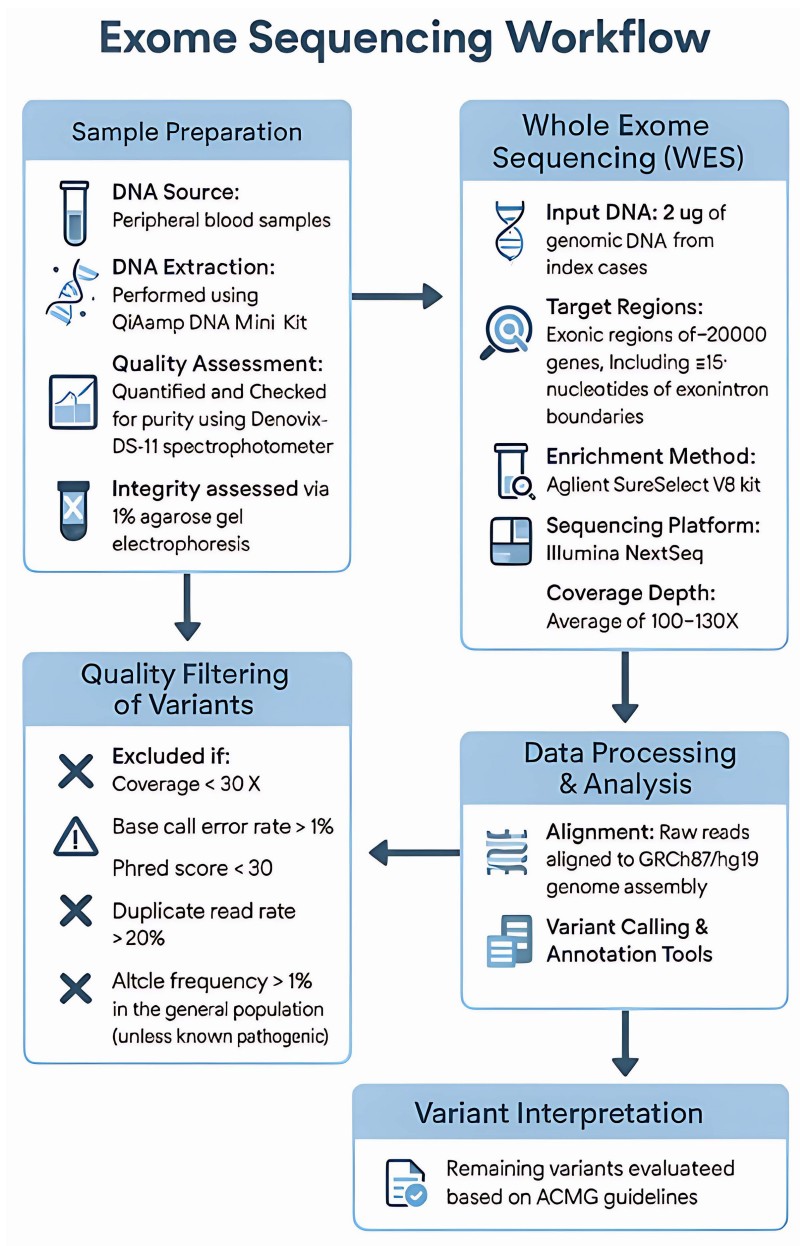

**Fig 1. Overview of the Exome Sequencing workflow.** Workflow includes sample preparation, sequencing using Illumina NextSeq, and variant filtering. Key quality control steps include DNA purity, coverage, and population frequency thresholds.

## 3. Results

### 3.1. Clinical phenotype and pedigree analysis

Sixteen patients from 12 different families were clinically diagnosed at the clinic. All patients were of Arab ethnicity and predominantly from Saudi Arabia, except for Families 8 and 12, who were Yemeni, and Family 11, who were Syrian, subsequently. However, all families currently reside in the Western region of Saudi Arabia.

Pedigree analysis suggests an autosomal recessive mode of inheritance in families, except for Family 10, which exhibits compound heterozygosity, and Family 12, which follows an autosomal dominant pattern (Fig 2). The probands in all families were born to consanguineous parents, except for Families 10 and 12.

The clinical features of all patients were consistent with typical phenotypes seen in DEB patients apart from two diagnosed with JEB. Detailed clinical patient information is provided in Table 1, with additional complications described in the S2 Table. Fig 3 shows Dystrophic EB characterized by marked skin fragility, with blisters forming in response to minimal trauma and typically healing with scarring. Nail abnormalities are commonly observed, ranging from dystrophic changes to the complete absence of fingernails and toenails. Repeated blistering and subsequent scarring can result in joint contractures and pseudosyndactyly fusion of the fingers and toes, further impacting mobility and hand function.

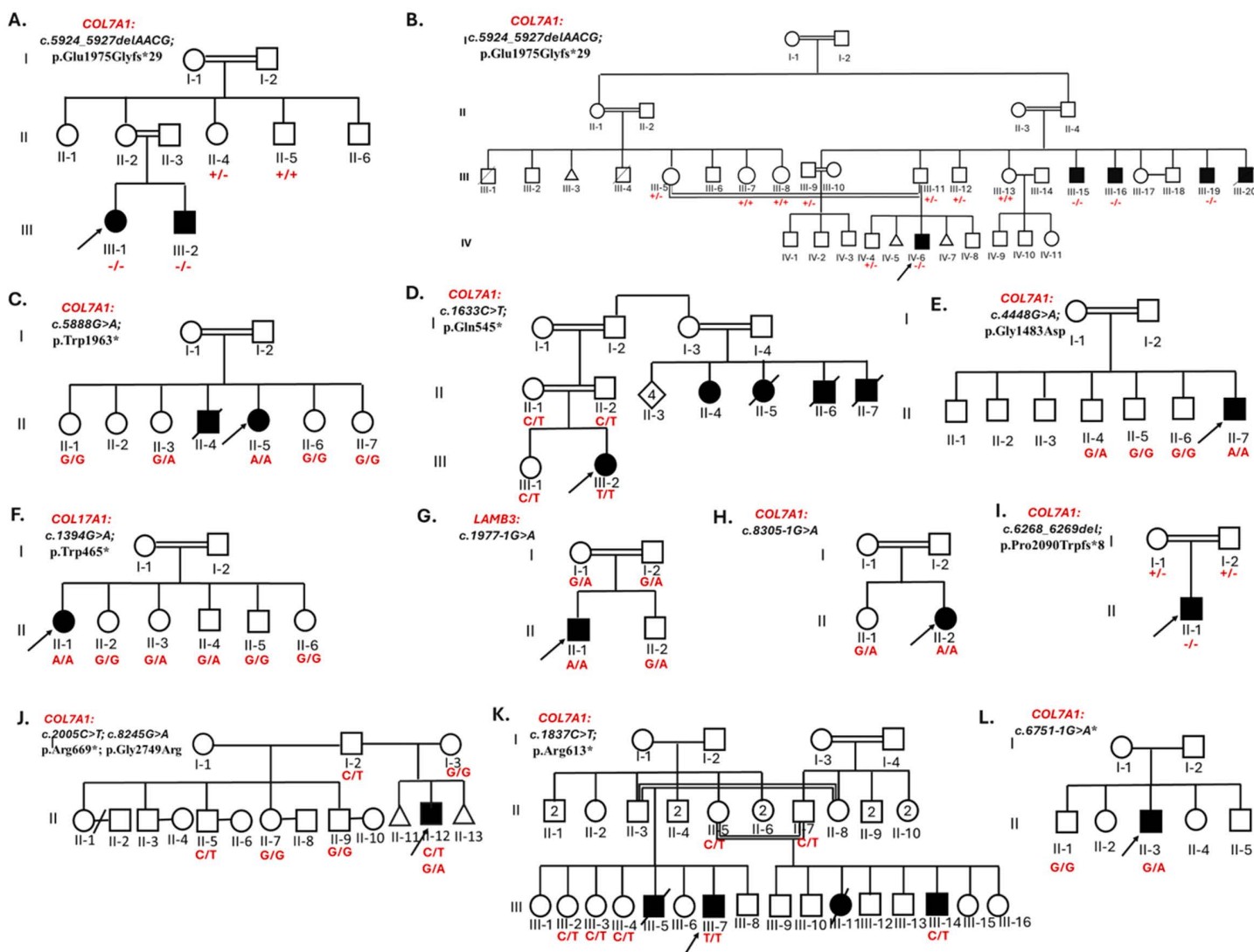

**Fig 2. Pedigrees of the 12 families with epidermolysis bullosa (EB) patients included in this study.** The arrow indicates the index case (proband) in each family. Panels A to L represent Families 1 to 12, respectively. Enrolled family members are shown with their corresponding genotype results. A black symbol indicates affected individuals. A square denotes a male, and a circle denotes a female.

**Table 1. Demographic details, clinical types, and symptoms of index cases.**

| Family ID | Patient ID | Age | Gender | Subtype | blisters | Pruritus | Nail dystrophy | Dental deformities | Ulcer in Oral | Alopecia |
|---|---|---|---|---|---|---|---|---|---|---|
| 1 | III-2 | 4 | Male | Dystrophic | Yes | Yes | Yes | Yes | Yes | Yes |
| 1 | III-1 | 8 | Female | Dystrophic | Yes | Yes | Yes | Yes | Yes | No |
| 2 | IV-6 | 9 | Male | Dystrophic | Yes | Yes | Yes | Yes | Yes | No |
| 2 | III-15 | 30 | Male | Dystrophic | Yes | Yes | Yes | Yes | Yes | No |
| 2 | III-16 | 29 | Male | Dystrophic | Yes | Yes | Yes | Yes | Yes | No |
| 2 | III-19 | 22 | Male | Dystrophic | Yes | Yes | Yes | Yes | Yes | No |
| 3 | II-5 | 27 | Female | Dystrophic | Yes | Yes | Yes | Yes | Yes | Yes |
| 4 | III-2 | 4 | Female | Dystrophic | Yes | Yes | Yes | Yes | Yes | No |
| 5 | II-7 | 1 | Male | Dystrophic | Yes | Yes | No | No | No | No |
| 6 | II-1 | 30 | Female | Junctional | Yes | Yes | Yes | Yes | Yes | Yes |
| 7 | II-1 | 6 | Male | Junctional | Yes | Yes | Yes | Yes | No | Yes |
| 8 | II-2 | 7 | Female | Dystrophic | Yes | Yes | Yes | Yes | No | No |
| 9 | II-1 | 1 | Male | Dystrophic | Yes | Yes | Yes | Yes | Yes | No |
| 10 | II-4 | 4 | Male | Dystrophic | Yes | Yes | Yes | Yes | Yes | No |
| 11 | III-7 | 7 | Male | Dystrophic | Yes | Yes | Yes | Yes | Yes | No |
| 12 | II-3 | 18 | Male | Dystrophic | Yes | Yes | Yes | Yes | Yes | No |

### 3.2. Genetic analysis

Whole-exome sequencing (WES) identified numerous variants in the probands from families based on the GRCh37/hg19 human genome assembly. Pathogenic and likely pathogenic variants in *COL7A1* (NM_000094.4) were identified in Families 1-5 and 8-12. Family 6 and Family 7 were affected by variants in *COL17A1* (NM_000494.4) and *LAMB3* (NM_000228.3), respectively. All identified variants with an autosomal recessive form of EB, except the c.6751-1G>A variant in *COL7A1*, which appeared to cause an autosomal dominant form of EB in Family 12. However, genetic testing of the parents declined, preventing confirmation of whether this variant arose de novo. In family 10, the c.2005C>T (p.Arg669*) and c.8245G>A (p.Gly2749Arg) variants were found in compound heterozygosity in the affected individual. In Family 6, a nonsense variant c.1394G>A (p.Trp465*) was identified in the COL17A1 gene (NM_000494.4), consistent with junctional epidermolysis bullosa. Notably, the *c.1633C>T* (p.Gln545*), *c.5888G>A* (p.Trp1963*), and *c.6268_6269del* (p.Pro2090Trpfs*8) variants in *COL7A1* are novel. Variant details are presented in Table 2. All findings were confirmed by Sanger sequencing, followed by family-based segregation analysis involving 59 individuals from different generations. The Sanger sequencing chromatogram is shown in Fig 4, and the segregation/genotyping results are summarized in the S3 Table. A review of previously reported variants is provided in Table 3.

### 3.3. Computational analysis of the variants' pathogenicity

The pathogenicity of the 12 variants identified in the *COL7A1*, *COL17A1*, and *LAMB3* genes was investigated using bioinformatics prediction tools, CADD, FATHMM-MKL, and SpliceAI. Ten of these variants were analyzed in detail. The CADD scores for these variants ranged from 23.7 to 45, all exceeding the pathogenicity threshold of 20, including a high likelihood of deleterious effects. Among these, only the missense variants showed FATHMM-MKL scores exceeding 0.99, suggesting significant impacts on protein function. The splice-site variant demonstrated a SpliceAI score close to 1.00, indicating a strong likelihood of disrupting normal splicing mechanisms. The identified variants included missense variants, stop-gain variants, splice site variants, and frameshift indels. The missense variants in *COL7A1*(NM_000094.4), c.8245G>A (p.Gly2749Arg) and c.4448G>A (p.Gly1483Asp), exhibited high pathogenicity scores across both CADD and FATHMM-MKL, suggesting a detrimental effect on protein structure and function. The stop-gain variants introduced

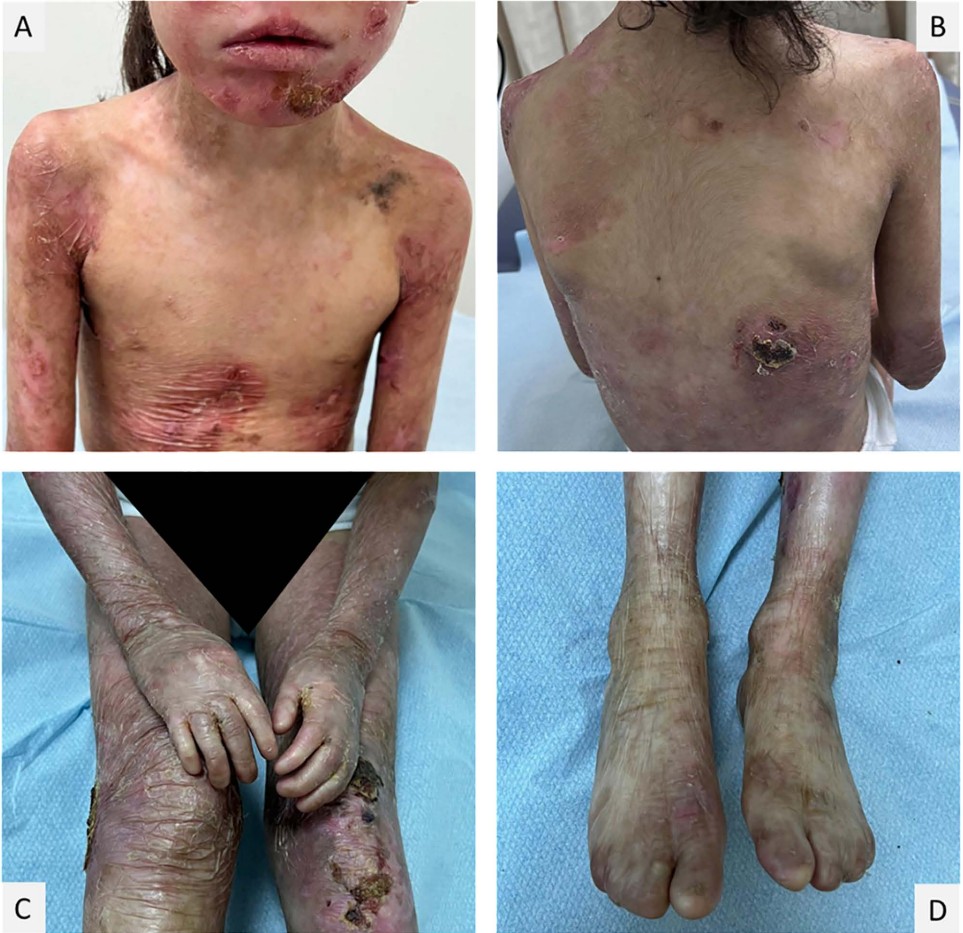

**Fig 3. A 7-year-old girl presenting dystrophic EB.** A. Blisters and erosions in the upper chest and lower part of the mouth causing difficulty with opening the mouth. B. crusts and atrophic scars. C. Thin, fragile, and translucent skin in the extremities. D. Repeated blistering and scarring of hands and feet cause fusion and webbing (pseudosyndacty) of the fingers and toes and absent nails.

premature stop codons, likely resulting in truncated, non-functional proteins, as supported by their high CADD scores (ranging from 33 to 41). All prediction scores are summarized in Table 4.

Splice site variants in both COL7A1 and LAMB3 genes showed SpliceAI scores of 0.99 or higher, indicating a high probability of disrupting normal splicing processes, as shown in Fig 5. Such disruptions can lead to aberrant mRNA transcripts and, subsequently, defective proteins. Two frameshift variants appeared in *COL7A1* (NM_000094.4); that is, c.5924_5927delAACG (p.Glu1975Glyfs*29) and c.6268_6269del (p.Pro2090Trpfs*8), both of which could not be assessed by using any in silico prediction tools due to limitations in scoring indels. In contrast, frameshift variants have been reported to change the gene's reading frame, leading to significant changes in the amino acid sequence downstream of the variant site and usually resulting in the synthesis of non-functional protein molecules. Despite the absence of predictive scores, these frameshift variants are considered clinically significant based on their likely impact on protein function. Collectively, the analysis suggests that all 12 variants have a high potential for pathogenicity. The consistency across multiple prediction tools—CADD, FATHMM-MKL, and Splice AI strengthens the evidence that these variants contribute to disease phenotypes associated with *COL7A1* and *LAMB3*. The high scores across these tools highlight the likely detrimental effects of the variants on protein function and RNA splicing, reinforcing their clinical relevance. The

**Table 2. Detailed annotations of the identified variants.**

| Family ID | Gene name | Genomic coordinates | Transcript | cDNA change | Amino acid | Allele frequency (gnomAD) | ACMG classification | dbSNP | Exon Reference |
|---|---|---|---|---|---|---|---|---|---|
| 1 | COL7A1 | Chr3:g.48613113_48613116del | NM_000094.4 | c.5924_5927del | p. Glu1975Glyfs*29 | 0% | Pathogenic | rs1064797080 | PMID:29531004 [26] |
| 2 | COL7A1 | Chr3:g.48613113_48613116del | NM_000094.4 | c.5924_5927del | p. Glu1975Glyfs*29 | 0% | Pathogenic | rs1064797080 | PMID:29531004 [26] |
| 3 | COL7A1 | Chr3:g.48613150C>T | NM_000094.4 | c.5888G>A | p. Trp1963* | 0% | Pathogenic | – | This study |
| 4 | COL7A1 | Chr3:g.48628900G>A | NM_000094.4 | c.1633C>T | p. Gln545* | 0% | Pathogenic | – | This study |
| 5 | COL7A1 | Chr3:g.48620594C>T | NM_000094.4 | c.4448G>A | p. Gly1483Asp | 0.001% | Pathogenic | rs756217590 | PMID:19197535 [27] |
| 6 | COL17A1 | Chr10:g.105816804C>T | NM_000494.4 | c.1394G>A | p. Trp465* | 0% | Pathogenic | – | PMID:16473856 [28] |
| 7 | LAMB3 | Chr1:g.209797346C>T | NM_000228.3 | c.1977–1G>A | – | 0% | Likely Pathogenic | rs786205451 | PMID:27124789 [29] |
| 8 | COL7A1 | Chr3:g.48603997C>T | NM_000494.4 | c.8305-1G>A | – | 0% | Likely Pathogenic | – | PMID:25877244 [30] |
| 9 | COL7A1 | Chr3:g.48612510_48612511del | NM_000094.4 | c.6268_6269del | p. Pro2090Trpfs*8 | 0% | Pathogenic | – | This study |
| 10 | COL7A1 | Chr3:48604152C>T | NM_000094.4 | c.2005C>T | p. Arg669* | 0.002% | Pathogenic | rs780261665 | PMID:16271705 [31] |
| | | Chr3:48566719G>A | | c.8245G>A | p.Gly2749Arg | 0.001% | Pathogenic | rs121912853 | PMID: 8644729 [32] |
| 11 | COL7A1 | Ch3:g.48627961C>T | NM_000094.4 | c.1837C>T | p.Arg613* | .002% | Pathogenic | rs759634066 | PMID:16971478 [33] |
| 12 | COL7A1 | Chr3:48610376C>T | NM_000094.4 | c.6751-1G>A | – | 0% | Pathogenic | – | PMID: 24213372 [34] |

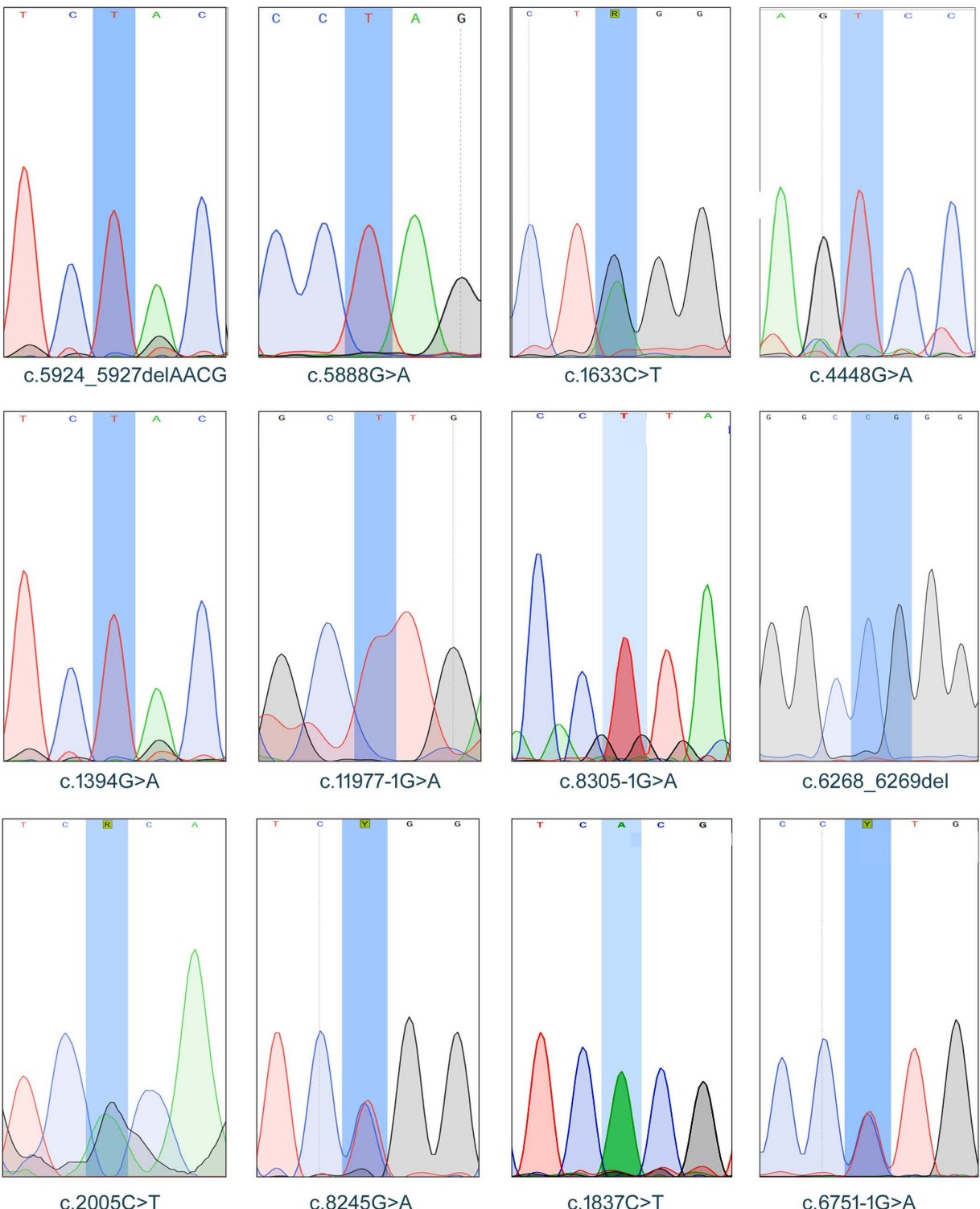

**Fig 4. Sanger sequencing chromatograms of pathogenic variants identified in the COL7A1, COL17A1, and LAMB3 genes among patients diagnosed with epidermolysis bullosa (EB).** The highlighted regions indicate the position of each variant. Reverse complement sequences are displayed where applicable.

**Table 3. The molecular and clinical details of variants identified in EB patients.**

| Variant ID | Population | #Patients | Zygosity | Clinical phenotype | Reference |
|---|---|---|---|---|---|
| *COL7A1* c.5924_5927delAACG | China | 1 | Homozygous | RDEB | [26] |
| | USA | 10 | Homozygous | DEB | [35] |
| *LAMB3* c.1977-1G>A | Europe | 1 | Homozygous | JEB | [36] |
| | Saudi Arabia | 1 | Homozygous | JEB | [6] |
| | Saudi Arabia | 2 | Homozygous | NH-JEB | [37] |
| *COL7A1* c.4448G>A | France | 3 | Homozygous | RDEB | [38] |
| | Kuwait | 2 | Homo-Hetero | RDEB-DEP | [39] |
| | Netherlands | 2 | Homozygous | RDEB | [40] |
| | Saudi Arabia | 1 | Homozygous | DEB | [6] |
| *COL7A1* c.8305-1G>A | China | 1- Parents Carrier | NA | DEB | [30] |
| *COL7A1* c.2005C>T | Chinese American | 1 | Compound Heterozygous | RDEB | [41] |
| | Switzerland | 1 | Compound Heterozygous | RDEB | [41] |
| | Dutch | 3 | NA | RDEB-sev | [42] |
| | Indonesia. | 1 | NA | RDEB | [43] |
| | Romania | 1 | NA | RDEB | [44] |
| COL7A1 c.8245G>A | Mexico | 1 | Homozygous | RDEB | [45] |
| | NA-SA | 1 | Homozygous | RDEB | [46] |
| *COL7A1* c.1837C>T | German | 1 | Compound Heterozygous | RDEB-sev | [47] |
| | Netherlands | 1 | Compound Heterozygous | RDEB | [48] |
| | NA | NA | NA | RDEB | [49] |
| | USA | 2 | Compound Heterozygous | RDEB | [50] |
| *COL7A1* c.6268_6269del | NA | NA | NA | NA | **Unreported** |
| *COL7A1* c.6751-1G>A | NA | NA | NA | NA | |
| *COL7A1* c.1633C>T | NA | NA | NA | NA | |
| *COL7A 1* c.5888G>A | NA | NA | NA | NA | |
| *COL17A1* c.1394G>A | NA | NA | NA | NA | |

splice site variants identified in the LAMB3 and COL7A1 genes (c.1977-1G>A, c.8305-1G>A, and c.6751-1G>A) meet multiple criteria under the ACMG/AMP guidelines for classification as pathogenic. All three are located at canonical splice acceptor sites (±1 or 2 positions) and thus fulfill the PVS1 (Very Strong) criterion, as loss-of-function (LOF) is an established mechanism of disease for both genes. These variants are absent or extremely rare in population databases such as gnomAD, satisfying the PM2 (Moderate) criterion. Furthermore, in silico predictions using tools like Splice AI strongly support their disruptive impact on normal splicing, providing evidence for PP3 (Supporting). These combined lines of evidence support the classification of all three variants as pathogenic, consistent with their expected consequences, such as exon skipping, intron retention, frameshifts, and premature stop codons that are likely to lead to non-functional or truncated proteins.

**Table 4. Details of the pathogenic prediction scores.**

| Gene | c.DNA | Amino Acid | Consequences | Pathogenic Prediction | | |
|------|-------|-----------|--------------|------|---------|----------|
| | | | | CADD | FATHAMM | Splice AI |
| *LAMB3* (NM000228.3) | c.1977-1G>A | – | splice acceptor | 34 | – | 0.99 (AL) |
| *COL7A1* (NM000094.4) | c.8245G>A | p.Gly2749Arg | Missense | 45 | 0.9953 | NA |
| *COL7A1* (NM000094.4) | c.4448G>A | p.Gly1483Asp | Missense | 23.7 | 0.9934 | NA |
| *COL7A1* (NM000094.4) | c.2005C>T | p.Arg669* | Stop gain | 33 | NA | NA |
| *COL7A1* (NM000094.4) | c.1837C>T | p.Arg613* | Stop gain | 34 | NA | NA |
| *COL7A1* (NM000094.4) | c.1633C>T | p.Gln545* | Stop gain | 37 | NA | NA |
| *COL17A1* (NM000494.4) | c.1394G>A | p.Trp465* | Stop gain | 41 | NA | NA |
| *COL7A1* (NM000094.4) | c.6751-1G>A | – | splice acceptor | 35 | NA | 1 (AL) |
| *COL7A1* (NM000094.4) | c.5888G>A | p.Trp1963* | Stop gain | 40 | NA | NA |
| *COL7A1* (NM000094.4) | c.8305-1G>A | – | splice acceptor | 33 | NA | 1(AL) |
| *COL7A1* (NM000094.4) | c.5924_5927del | p.Glu1975Glyfs*29 | Frameshift | NA | NA | NA |
| *COL7A1* (NM000094.4) | c.6268_6269del | p.Pro2090Trpfs*8 | Frameshift | NA | NA | NA |

CADD = Combined Annotation Dependent Depletion; FATHMM = Functional Analysis Through Hidden Markov Models; NA = Not Available; SpliceAI = Splicing prediction score, with "AL" indicating Acceptor Loss. Amino acid changes are described according to HGVS nomenclature based on transcript NM_000094.4 for COL7A1 and NM_000228.3 for LAMB3.

## 3.4. Splice site variants and RNA secondary structure analysis

For the *LAMB3* (NM000228.3) gene, the splice acceptor variant c.1977-1G>A was analyzed using RNAfold to interpret the effect of the variant on the RNA secondary structure. The minimum free energy (MFE) for the wild-type RNA was calculated to be −132.75 kcal/mol, whereas the mutant RNA structure exhibited an MFE of −126.52 kcal/mol, showing that the RNA became less stable due to the variant. Additionally, there is a shift in the free energy of the thermodynamic ensemble from −140.95 kcal/mol for the wild type to −136.10 kcal/mol for the mutant. All these changes reflect a less stable RNA structure in the mutant, which might affect RNA folding and splicing efficiency. As a result, disruption of standard structural elements in mutant RNA may disturb usual RNA processing with subsequent impaired gene expression (Fig 6).

This study investigated two *COL7A1*(NM000094.4) gene splice site variants: c.6751-1G>A and c.8305-1G>A. In the case of variant c.6751-1G>A, the MFE of wild-type RNA was -102.38 kcal/mol while that of the mutant RNA was -96.75 kcal/mol, thus pointing toward the destabilization of the RNA structure in the mutant. In the thermodynamic ensemble, the free energy also increased from -109.45 kcal/mol for the wild type to -103.10 kcal/mol for the mutant, thus further indicating reduced stability of RNA. In the case of the c.8305-1G>A variant, the MFE of wild-type RNA was -116.85 kcal/mol versus -111.45 kcal/mol for the mutant, thus indicating stability reduction. Similarly, the free energies of the thermodynamic ensemble increased from -122.30 kcal/mol to -117.35 kcal/mol in the mutant version. Both variants indicate that the variants result in less stable RNA folds, which may affect normal splicing and RNA function (Fig 7).

 

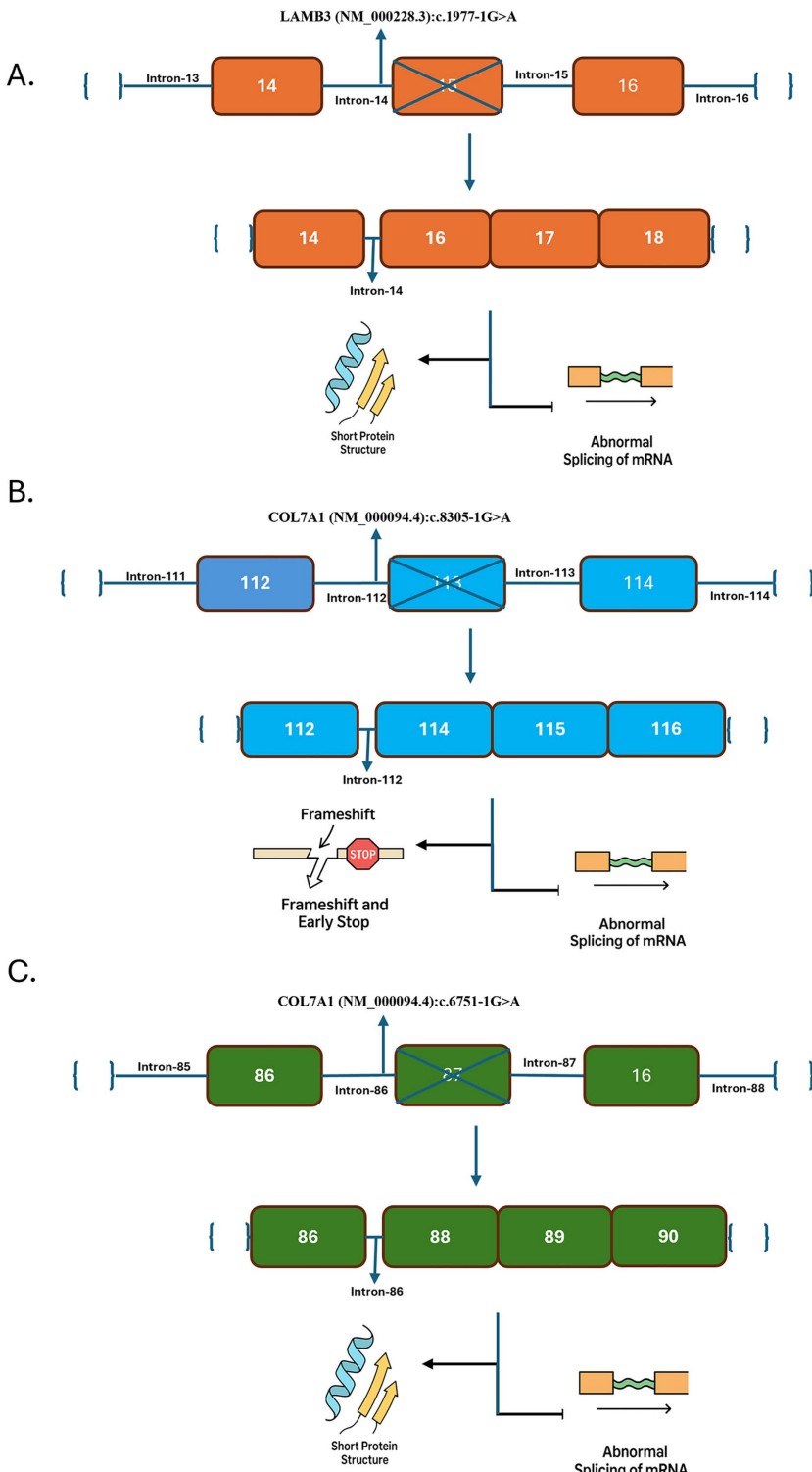

**Fig 5. Splice site variants in (a) LAMB3 c.1977-1G>A, (b) COL7A1 c.8305-1G>A, and (c) COL7A1 c.6751-1G>A are shown with exons and introns.** Each variant disrupts the respective splice acceptor site, leading to exon skipping, intron retention, or frameshift. These events likely result in premature stop codons and truncated proteins.

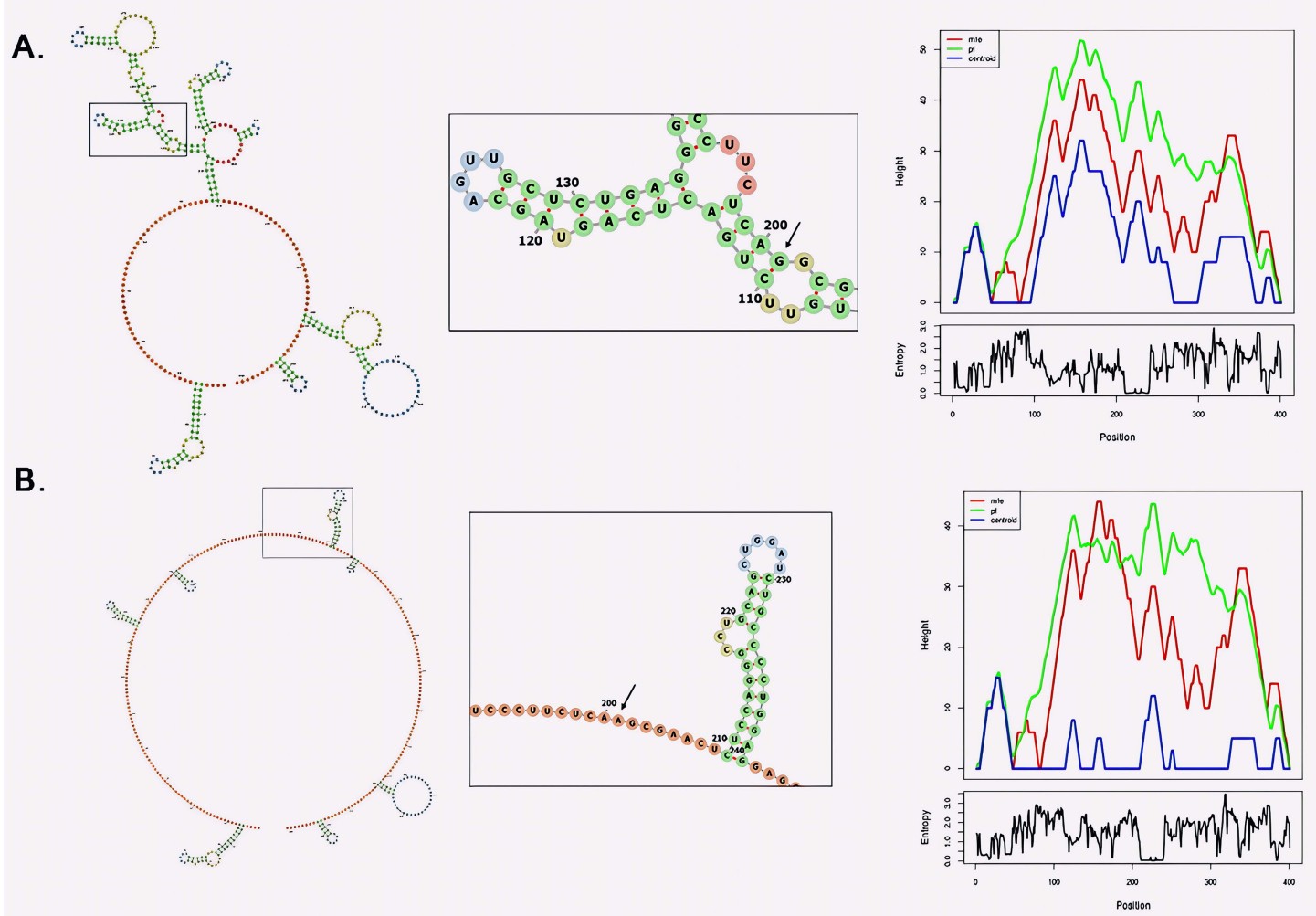

**Fig 6. RNA secondary structure and mountain plot for *LAMB3* variants identified in EB patients (A) Wild-type RNA structure of the *LAMB3* gene at the c.1977-1G>A splice acceptor site, showing a minimum free energy (MFE) of −132.75 kcal/mol.** (B) Mutant RNA structure of the *LAMB3* gene after the c.1977-1G>A variant, with an MFE of −126.52 kcal/mol, indicates a stability decrease. The mountain plot comparison highlights structural changes between the wild-type and mutant RNA folds.

### 3.5. Protein structural analysis

Protein models for both the wild-type and mutant forms of the *COL7A1* gene were developed using the AlphaFold2 tool, covering the full-length protein, which consists of 2,944 amino acids (Fig 8). This represents the first complete model of this complex protein, as no experimental structure was previously available. The variants p. Gly1483Asp and p. Gly2749Arg were analyzed to assess their structural impact. Accordingly, Root Mean Square Deviation (RMSD) values were computed for both variants: in p. Gly1483Asp, RMSD reaches a value of 3.5 Å, while in p. Gly2749Arg, it has a value of 3.8 Å. This relatively high RMSD may indicate that the variant introduces significant deviations in the overall protein structure. Given the large size and complexity of *COL7A1*, such structural alterations will likely affect critical domains, potentially impairing protein function. This work represents a significant advancement in the structural understanding of *COL7A1* and provides a valuable foundation for future studies on its biological function and role in disease.

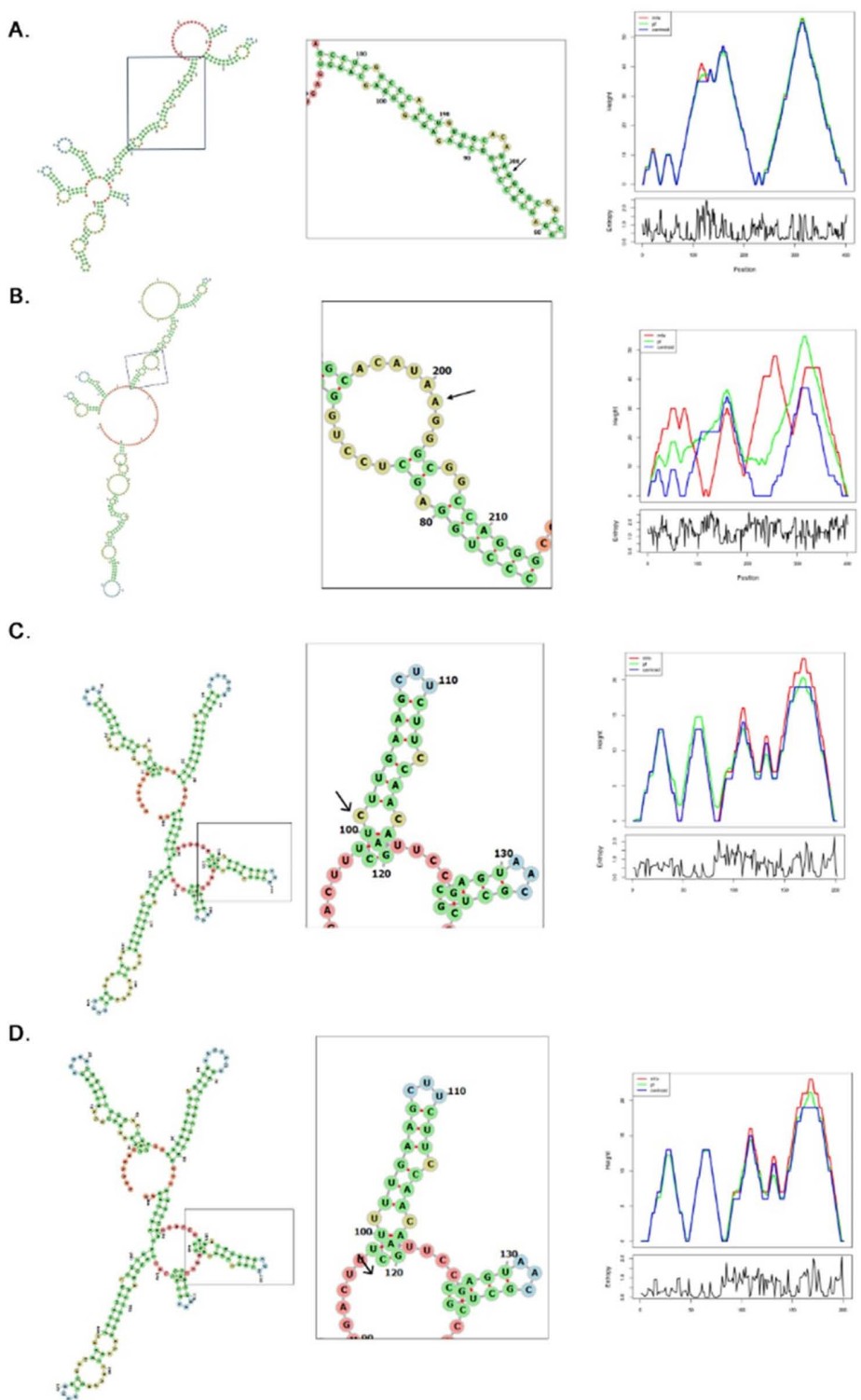

**Fig 7. RNA secondary structure analysis for two splice site variants in *COL7A1*.** (A) Wild-type RNA structure for the c.6751-1G>A variant with an MFE of −102.38 kcal/mol. (B) Mutant RNA structure for the c.6751-1G>A variant showing an MFE of −96.75 kcal/mol, suggesting destabilization. (C) Wild-type RNA structure for the c.8305-1G>A variant with an MFE of −116.85 kcal/mol. (D) Mutant RNA structure for the c.8305-1G>A variant showing an MFE of −111.45 kcal/mol, indicating reduced RNA stability. The mountain plots further illustrate the structural changes in both variants, reflecting the impact of the variants on RNA folding.

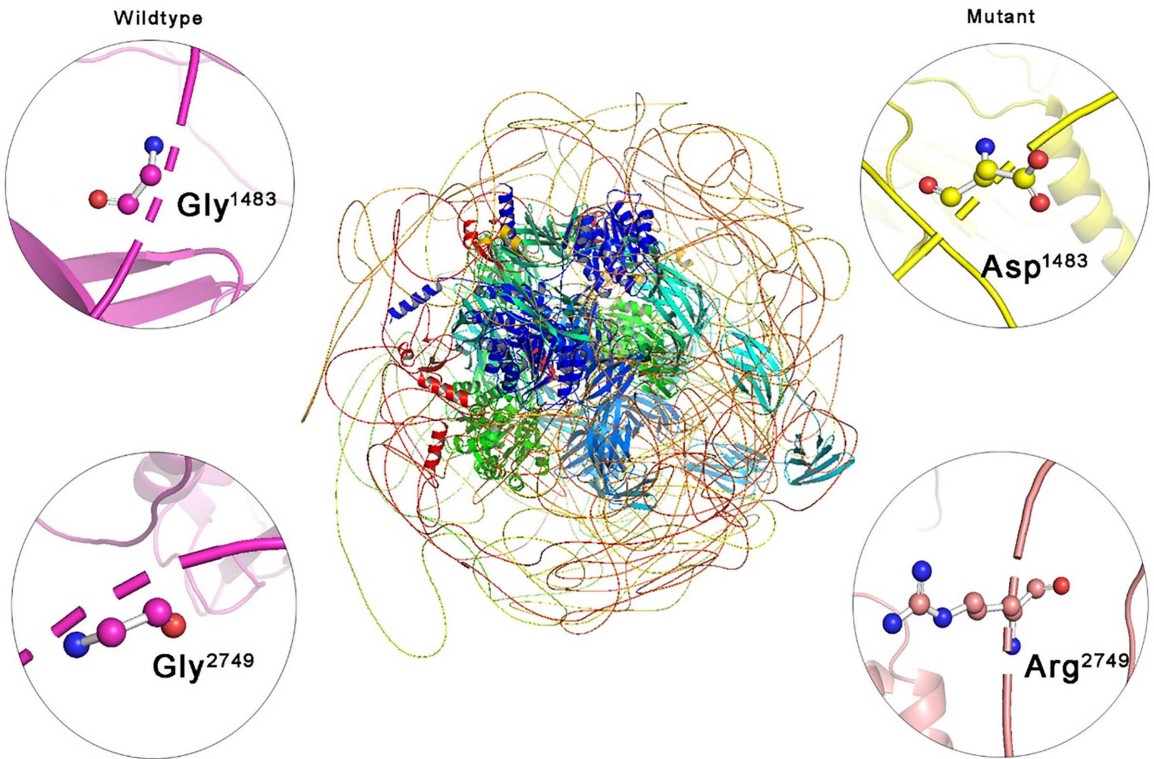

**Fig 8. Structural models of the *COL7A1* protein (2,944 amino acids) generated using AlphaFold2, showing wild-type and mutant structures for the p.Gly1483Asp and p.Gly2749Arg variants.** The central image represents full-length protein, with close-up views highlighting the wild-type Gly1483 and Gly2749 residues (left) and their mutated counterparts, Asp1483 and Arg2749 (right).

## 4. Discussion

The current study highlights the genetic makeup of Middle Eastern Arab EB patients by detecting causative variants in the *COL7A1*, *COL17A1*, and *LAMB3* genes. Notably, the discovery of five previously unreported variants highlights the unique genetic characteristics of EB within consanguineous families. This work builds upon previous efforts to elucidate the molecular basis of EB in the region [11]. In addition to expanding the known spectrum of EB variants, this study underscores the importance of region-specific genetic profiling. Such profiling is essential for understanding genotypes and phenotype correlations, particularly in populations with a high rate of consanguinity.

The *COL7A1* gene codes for type VII collagen, a critical component in anchoring fibrils that link the dermal and epidermal layers of the skin [51]. Pathogenic variants in *COL7A1* are well-documented as the cause of dystrophic epidermolysis bullosa (DEB), with recessive inheritance typically associated with more severe clinical presentations, particularly in consanguineous populations [52]. In this study, a total of ten variants were identified in *COL7A1*(NM000094.4), including two missense variants c.4448G>A (p.Gly1483Asp) and c.8245G>A (p. Gly2749Arg), two frameshift variants c.5924_5927delAACG (p.Glu1975Glyfs*29) and c.6268_6269del (p.Pro2090Trpfs*8), two splice site variants (c.8305-1G>A, c.6751-1G>A), and four truncating variants c.1633C>T (p.Gln545*), c.2005C>T(p.Arg669*), c.1837C>T (p. Arg613*) and c.5888G>A (p.Trp1963*). Of these *COL7A1* variants, c.1633C>T (p. Gln545*), c.5888G>A (p. Trp1963*), and c.8305-1G>A were novel. These variants, novel to the Saudi population, reveal region-specific genetic changes in DEB and illustrate distinct mechanisms by which type VII collagen dysfunction contributes to EB pathology. These are novel variants in the Saudi population, reflecting region-specific genetic changes in DEB and showing distinct

mechanisms whereby dysfunction in type VII collagen contributes to the pathology of EB. The identified variants disrupt the synthesis or function of type VII collagen, weakening anchoring fibrils and ultimately compromising dermal-epidermal adhesion, thereby contributing to the clinical manifestations of EB [53].

Missense variants that substitute glycine residues within the triple-helix domain of type VII collagen destabilize the structure of collagen and weaken fibril stability. In our cohort, two notable missense variants were identified: c.4448G>A (p.Gly1483Asp) and c.8245G>A (p.Gly2749Arg). The c.4448G>A variant replaces glycine with aspartic acid, whereas the c.8245G>A variant replaces glycine with arginine. Glycine is a critical component of the collagen triple helix due to its small size, which permits tight packing and stability. Substituting glycine with larger residues introduces steric hindrance, disrupts the helical structure, reduces flexibility, and weakens anchoring fibrils. Moreover, the bulky and charged nature of aspartic acid and arginine interferes with fibril stability and dermal-epidermal adhesion, leading to the severe blistering and scarring that characterizes DEB patients [54]. The identification of p.Gly1483Asp and p.Gly2749Arg highlights the critical role of glycine residues in maintaining collagen integrity and stability and suggests that glycine substitutions are significant pathogenic drivers in DEB. The calculated RMSD values of 3.5 Å p.Gly1483Asp and 3.8 Å for p.Gly2749Arg reflect significant structural diversities when compared with the wild-type *COL7A1* protein, indicating significant changes in protein conformation. These alterations likely compromise collagen stability and function, contributing to the clinical severity of DEB and emphasizing the pathogenic potential of these variants within the Saudi population.

Several stop-gain variants in *COL7A1* and *COL17A1* are identified in our study; these disrupt protein structures critical to dermal-epidermal cohesion, leading to severe dystrophic epidermolysis bullosa (DEB) phenotypes [52]. Stop-Gain Variants in *COL7A1* (NM_000094.4): variants like c.1633C>T (p.Gln545*), c.2005C>T (p.Arg669*), c.5888G>A (p.Trp1963*), and c.1837C>T (p.Arg613*) introduce premature stop codons within the *COL7A1* sequence, leading to truncated protein products that are unable to assemble into functional anchoring fibrils. The variant p.Gln545* at exon 12 results in the loss of critical C-terminal domains required for collagen helix formation, thereby compromising fibril assembly. Similarly, the p.Arg669* variant in exon 15 leads to the synthesis of an incomplete type VII collagen protein lacking important regions for fibril stability and interaction with other structural proteins, including significant portions of the glycine-X-Y repeat regions [55]. The p.Trp1963* variant results in the loss of the C-terminal NC1 domain, which is crucial for collagen stability and proper fibril formation at the dermal-epidermal junction [53]. The p.Arg613* variant leads to a loss of portions of the triple-helical region and further downstream domains, affecting the protein's stability and ability to form functional fibrils. Additionally, The c.1394G>A (p.Trp465*) variant in *COL17A1* introduces a premature stop codon in type XVII collagen [13]. Such a variant results in a truncated protein, leading to impaired hemidesmosome formation at the dermal-epidermal junction and further fragility of the skin. Clinically, DEB is due to several of these kinds of variants, which provide less structural integrity, with the result of extensive blistering, chronic wounds, and scarring [52].

The frameshift variant c.6268_6269del (p.Pro2090Trpfs*8) disrupts the reading frame in *COL7A1*, resulting in altered downstream amino acids and introducing a premature stop codon. These variant impacts key domains necessary for collagen VII's structural integrity, particularly disrupting the triple-helical domain and the C-terminal NC1 domain, both essential for stable fibril assembly and dermal-epidermal cohesion. Frameshift variants like p.Pro2090Trpfs*8 typically trigger nonsense-mediated decay (NMD) or produce truncated, non-functional proteins lacking these critical regions. The resulting reduction in functional type VII collagen impairs anchoring fibril formation, a hallmark of severe dystrophic epidermolysis bullosa (DEB). Patients with frameshift variants in *COL7A1* often present with extensive blistering, scarring, and severely compromised tissue integrity, underscoring that even small deletions can profoundly disrupt collagen assembly and skin stability [56].

The splice-site variants identified in *COL7A1* (NM_000094.4), (c.6751-1G>A and c.8305-1G>A) and *LAMB3* (NM_000228.3), c.1977-1G>A are canonical splice acceptor site variants located at the intron-exon boundaries. These variants likely disrupt normal splicing by causing exon skipping or activating cryptic splice sites, resulting in aberrant mRNA transcripts. RNAfold analysis revealed increased minimum free energy (MFE) values in the mutant sequences,

indicating decreased RNA stability and suggesting impaired splicing efficiency. This reduced stability affects the production of functional type VII collagen and laminin-332, proteins essential for dermal-epidermal adhesion. In *COL7A1*, variants c.6751-1G>A and c.8305-1G>A are predicted to compromise proper RNA processing, ultimately impairing the synthesis of functional type VII collagen. This defect disrupts the formation of anchoring fibrils essential for dermal-epidermal cohesion, directly contributing to the blistering phenotype observed in dystrophic epidermolysis bullosa (DEB). The disruption of these anchoring fibrils is directly related to DEB's blistering phenotype. Similarly, the *LAMB3* splice-site variant c.1977-1G>A likely leads to improper splicing and reduced stability of the transcript, impairing the production of laminin-332 production [57]. Laminin-332 is typically essential for the stability of hemidesmosomes that anchor cells at the dermal-epidermal junction. Disruption in hemidesmosome formation due to deficient laminin-332 compromises cell adhesion, contributing to the severe blistering and poor wound healing seen in junctional epidermolysis bullosa (JEB) patients [58]. The presence of these splice-site variants highlights the molecular basis of EB, where RNA destabilization leads to protein deficiencies, which underlie tissue fragility and the characteristic EB pathology.

Several COL7A1 variants identified in our Saudi EB patients overlap with globally reported mutations. The c.5924_5927del (p.Glu1975Glyfs29) frameshift was previously seen in Chinese and U.S. patients with severe RDEB [26]. The c.4448G>A (p.Gly1483Asp) missense variant found in our cohort matches reports from France and the Netherlands (PMID: 19197535) [27], while its heterozygous form in Kuwait showed a milder phenotype (PMID: 8644729) [32]. The c.2005C>T (p.Arg669) nonsense variant, identified in a compound heterozygous state, has been reported in Chinese-American and Swiss cases with RDEB [31]. Other variants, such as c.8305-1G>A (splice-site) [30] and c.8245G>A (missense) [32] were consistent with published RDEB phenotypes. Three novel mutations; c.1633C>T, c.5888G>A, and c.6268_6269del were found exclusively in our cohort, yet their clinical presentations were comparable to classic DEB. A dominant splice variant, c.6751-1G>A, aligned with previous dominant DEB reports [34]. The LAMB3 variant c.1977-1G>A observed in JEB matched earlier Saudi and European cases [29]. These findings confirm both overlaps with global data and region-specific contributions to the EB mutation landscape.

## 5. Conclusion

This study expands the global understanding of EB by identifying novel variants in *COL7A1*, *COL17A1*, and *LAMB3* within the Middle Eastern Arab population. These findings underscore the need for region-specific genetic profiling to aid targeted diagnostic and therapeutic strategies in EB patients in Saudi Arabia and other populations with high rates of consanguinity. Establishing variant-specific databases and developing precision therapies will be crucial for improving patient outcomes and advancing EB research. Looking ahead, large-scale genetic studies across the Middle East could further refine variant databases and improve genetic counseling practices. By identifying unique genetic markers, such research could contribute to early genetic diagnosis, management, and treatment outcomes for EB. The findings from this study provide a foundation for future therapeutic development in EB, particularly in designing variant-specific therapies that address the unique molecular pathology observed in Arab EB patients.

## Supporting information

**S1 Table. Primer details used in PCR and Sanger sequencing.**
(DOCX)

**S2 Table. Clinical complications across multiple systems in index cases from Saudi EB families.**
(DOCX)

**S3 Table. Results of the segregation analysis for the identified variants in affected and unaffected family members.**
(DOCX)

## Acknowledgments

The authors would like to thank the Saudi Society for Laboratory Medicine for their scientific support.

## Author contributions

**Conceptualization:** Hadiah Bassam Al Mahdi, Shmoukh Alghuraibi.

**Data curation:** Sultana Abdulghani.

**Formal analysis:** Babajan Banaganapalli, Fahad Hakami.

**Funding acquisition:** Zuhier Awan.

**Investigation:** Nancy Shehata, Fahad Hakami.

**Methodology:** Babajan Banaganapalli.

**Project administration:** Hadiah Bassam Al Mahdi, Shmoukh Alghuraibi.

**Software:** Babajan Banaganapalli.

**Supervision:** Mahmoud Younis.

**Validation:** Hadiah Bassam Al Mahdi, Noor Ahmad Shaik.

**Writing – original draft:** Nancy Shehata, Hadiah Bassam Al Mahdi, Shmoukh Alghuraibi.

**Writing – review & editing:** Noor Ahmad Shaik, Zuhier Awan, Fahad Hakami.

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
