## [Decision Letter · Decision Letter 0]

2 Apr 2025

Dear Dr. Shehata.

Thank you for submitting your manuscript to PLOS ONE. After careful consideration, we feel that it has merit but does not fully meet PLOS ONE’s publication criteria as it currently stands. Therefore, we invite you to submit a revised version of the manuscript that addresses the points raised during the review process.

**The findings of the manuscript provide valuable insights on   molecular basis of Epidermolysis Bullosa (EB) from Middle eastern families. The study  reports novel and known pathogenic mutations in three genes (*COL7A1* , *COL17A1* , *LAMB3* )  from  12 families affected with EB by employing exome sequencing. The manuscript is well structured technically sound. However the manuscript requires minor reversions for improvements of quality and better comprehension before acceptance for publication. The following points needs to be considered:**

**As the study reports families from Saudi Arabia, Yemen and Syria, so the title referring ‘ Saudi Arabian families’ is misleading. The use of  term ‘Middle eastern’ would have been more appropriate.****The quality of pedigrees in  figure 1 needs considerable  improvements for clarity.**** A  flow chart  showing  exome sequencing pipeline will be useful.****More Information on inclusion  and diagnostic criteria, details on clinical phenotypes should be added.****Some gene mutation nomenclature  should be checked especially the variant described at protein level. Please fellow the guideline set by Human Genome Variation Society (HGVS).****Functional consequences of splice site and misssense variant should be  discussed in discussion.****There are grammatical and typological  errors in different sections of the manuscript which needs to corrected**

**Furthermore, the reviewers’ feedback provide extensive reversion points to consider that  will be useful in improving the quality of the manuscript.**

We look forward to receiving your revised manuscript.

Kind regards,

Rabia Habib, Ph.D

Academic Editor

PLOS ONE

**Journal Requirements:**

1. When submitting your revision, we need you to address these additional requirements. Please ensure that your manuscript meets PLOS ONE's style requirements, including those for file naming. The PLOS ONE style templates can be found at https://journals.plos.org/plosone/s/file?id=wjVg/PLOSOne_formatting_sample_main_body.pdf and https://journals.plos.org/plosone/s/file?id=ba62/PLOSOne_formatting_sample_title_authors_affiliations.pdf 2. We suggest you thoroughly copyedit your manuscript for language usage, spelling, and grammar. If you do not know anyone who can help you do this, you may wish to consider employing a professional scientific editing service.  The American Journal Experts (AJE) (https://www.aje.com/) is one such service that has extensive experience helping authors meet PLOS guidelines and can provide language editing, translation, manuscript formatting, and figure formatting to ensure your manuscript meets our submission guidelines. Please note that having the manuscript copyedited by AJE or any other editing services does not guarantee selection for peer review or acceptance for publication.  Upon resubmission, please provide the following: The name of the colleague or the details of the professional service that edited your manuscript A copy of your manuscript showing your changes by either highlighting them or using track changes (uploaded as a *supporting information* file) A clean copy of the edited manuscript (uploaded as the new *manuscript* file) 3. Please provide additional details regarding participant consent. In the ethics statement in the Methods and online submission information, please ensure that you have specified what type you obtained (for instance, written or verbal, and if verbal, how it was documented and witnessed). If your study included minors, state whether you obtained consent from parents or guardians. If the need for consent was waived by the ethics committee, please include this information. Once you have amended this/these statement(s) in the Methods section of the manuscript, please add the same text to the “Ethics Statement” field of the submission form (via “Edit Submission”). For additional information about PLOS ONE ethical requirements for human subjects research, please refer to http://journals.plos.org/plosone/s/submission-guidelines#loc-human-subjects-research. 4. Thank you for stating in your Funding Statement: This publication is based on work supported by The Saudi Society for Laboratory Medicine for funding the research.  Please provide an amended statement that declares *all* the funding or sources of support (whether external or internal to your organization) received during this study, as detailed online in our guide for authors at http://journals.plos.org/plosone/s/submit-now.  Please also include the statement “There was no additional external funding received for this study.” in your updated Funding Statement. Please include your amended Funding Statement within your cover letter. We will change the online submission form on your behalf. 5. Thank you for stating the following in the Acknowledgments Section of your manuscript: This publication is based on work supported by The Saudi Society for Laboratory Medicine for funding the research. We note that you have provided funding information that is not currently declared in your Funding Statement. However, funding information should not appear in the Acknowledgments section or other areas of your manuscript. We will only publish funding information present in the Funding Statement section of the online submission form. Please remove any funding-related text from the manuscript and let us know how you would like to update your Funding Statement. Currently, your Funding Statement reads as follows: This publication is based on work supported by The Saudi Society for Laboratory Medicine for funding the research.   Please include your amended statements within your cover letter; we will change the online submission form on your behalf. 6. We note that there is identifying data in the Supporting Information file “EB Supplementary Tables”. Due to the inclusion of these potentially identifying data, we have removed this file from your file inventory. Prior to sharing human research participant data, authors should consult with an ethics committee to ensure data are shared in accordance with participant consent and all applicable local laws. Data sharing should never compromise participant privacy. It is therefore not appropriate to publicly share personally identifiable data on human research participants. The following are examples of data that should not be shared: -Name, initials, physical address-Ages more specific than whole numbers-Internet protocol (IP) address-Specific dates (birth dates, death dates, examination dates, etc.)-Contact information such as phone number or email address-Location data-ID numbers that seem specific (long numbers, include initials, titled “Hospital ID”) rather than random (small numbers in numerical order) Data that are not directly identifying may also be inappropriate to share, as in combination they can become identifying. For example, data collected from a small group of participants, vulnerable populations, or private groups should not be shared if they involve indirect identifiers (such as sex, ethnicity, location, etc.) that may risk the identification of study participants. Additional guidance on preparing raw data for publication can be found in our Data Policy (https://journals.plos.org/plosone/s/data-availability#loc-human-research-participant-data-and-other-sensitive-data) and in the following article: http://www.bmj.com/content/340/bmj.c181.long. Please remove or anonymize all personal information (<specific identifying information in file to be removed>), ensure that the data shared are in accordance with participant consent, and re-upload a fully anonymized data set. Please note that spreadsheet columns with personal information must be removed and not hidden as all hidden columns will appear in the published file. 7. PLOS requires an ORCID iD for the corresponding author in Editorial Manager on papers submitted after December 6th, 2016. Please ensure that you have an ORCID iD and that it is validated in Editorial Manager. To do this, go to ‘Update my Information’ (in the upper left-hand corner of the main menu), and click on the Fetch/Validate link next to the ORCID field. This will take you to the ORCID site and allow you to create a new iD or authenticate a pre-existing iD in Editorial Manager. 8. Your ethics statement should only appear in the Methods section of your manuscript. If your ethics statement is written in any section besides the Methods, please delete it from any other section. 9. Please upload a new copy of Figure 1 as the detail is not clear. Please follow the link for more information: https://blogs.plos.org/plos/2019/06/looking-good-tips-for-creating-your-plos-figures-graphics/" https://blogs.plos.org/plos/2019/06/looking-good-tips-for-creating-your-plos-figures-graphics/

Reviewers' comments:

Reviewer's Responses to Questions

**Comments to the Author**

1. Is the manuscript technically sound, and do the data support the conclusions?

Reviewer #1: Yes

Reviewer #2: Yes

Reviewer #3: Partly

2. Has the statistical analysis been performed appropriately and rigorously?

Reviewer #1: N/A

Reviewer #2: Yes

Reviewer #3: Yes

3. Have the authors made all data underlying the findings in their manuscript fully available?

Reviewer #1: Yes

Reviewer #2: Yes

Reviewer #3: Yes

4. Is the manuscript presented in an intelligible fashion and written in standard English?

Reviewer #1: Yes

Reviewer #2: Yes

Reviewer #3: No

**Reviewer #1: ** This manuscript will make a nice contribution to the field of human genetics. But needs some improvements and minor revisions before formal acceptance for publication in PlosOne. Authors should considering following points during minor revisions:

1. Authors must provide the transcript IDs of different genes in which mutations have been identified.

2. In the abstract there are empty parentheses "()".

3. Quality of all the figures needs improvement.

4. A figure detailing the flow chart of exome sequencing should be made for clear understanding.

5. There are some splice variants, these splice variants should be analyzed using bioinformatics softwares like NetGene2 etc. for assessing the effect of these variants. And graphical representation in the form of a a figure showing intron inclusion in the final protein or exon skipping due to aberrant splicing should be given.

6. Carefully check the grammar and other typographical mistakes before resubmission.

**Reviewer #2: ** The study presents significant findings on novel and recurrent mutations in EB among Saudi families, contributing valuable data on a region with limited molecular epidemiological studies. The manuscript is well-structured and follows a logical flow, but some sections require refinement for clarity and conciseness.

Specific Comments:

Title & Abstract

- The title is appropriate but could be more specific by highlighting "molecular findings" or "genetic profiling."

- The abstract summarizes key findings well, but some sentences are wordy. Consider shortening complex phrases for better readability.

- The Results section should clearly differentiate between novel and recurrent mutations in the abstract.

- The conclusion should explicitly state the clinical significance of the study.

Introduction

- The introduction provides good background information, but the prevalence statistics on EB in Saudi Arabia should be expanded with references.

- The final paragraph should clearly define the research gap and justify why studying Saudi families is important.

Methodology

- Recruitment Criteria: The inclusion and exclusion criteria should be more explicit, especially concerning how patients were diagnosed and classified.

- Computational Analysis: Provide references for the specific thresholds used in pathogenicity predictions (e.g., CADD score cutoff).

Results

- Clearly separate findings related to novel versus known variants to emphasize new contributions.

- Consider reorganizing the section to first present clinical findings, then genetic results, and finally computational predictions.

Discussion

- Comparisons with global EB cases are insightful, but more emphasis should be placed on how Saudi genetic variants differ from other populations.

- Implications for genetic counseling in consanguineous populations should be elaborated on.

Conclusion

- The conclusion should be more concise and explicitly highlight future research directions.

- Mention potential applications of findings, such as early genetic screening programs.

Figures & Tables

- Figure 1 (Pedigrees): Improve clarity by ensuring consistent labeling across all families.

- Figure 2 (Clinical Presentation): Enhance image descriptions to clarify key features of EB.

- Tables 3 & 4: Consider moving some detailed variant data to supplementary material to improve readability.

**Reviewer #3: ** The authors have done a good work on the EB families from the middle east. But I have some suggestions and changes for the integrity of the manuscript, I trust in the authors’ capabilities to address all the comments.

# This study includes families from three ethnic origins i.e., Saudi Arabia, Syria and Yemen, so how the authors focus in the title only on the Saudi Arabian families?

# “This study intends to shed some light on recurrent as well as novel genetic abnormalities that cause EB in the Western region of Saudi Arabia” is a poor presentation please rephrase this sentence e.g., “This study intends to highlight the recurrent as well as novel genetic abnormalities that cause EB in the Western region of Saudi Arabia”.

# “Twelve EB families in Saudi Arabia were recruited from clinical settings” this sentence is ambiguous to understand please try to make its sense clearer. What do you mean by clinical settings here?

# In line 43 (abstract), please correct “classify disease subtypes precisely” as “classified the disease subtypes precisely”.

# In line 44 “to identify EB-causing variants” as “to identify EB-causing gene variants”.

# In line 47, the authors state that “We identified 11 pathogenic variants”. Were all the variants pathogenic, confirmed by the pathogenicity prediction tools? Otherwise rephrase this sentence and use correct terms regarding classification of the identified variants.

# Use correct nomenclature regarding the identified frameshift, nonsense, and missense variants nonsense (c.1633C>T, c.1837C>T, c.2005C>T, and c.5888G>A), missense (c.4448G>A, and c.8245G>A). Please follow the variant nomenclature guidelines, how to write a variant that has not been verified through functional studies in the model organisms.

# The sentences “The COL7A1 variants included frameshift (c.5924_5927del and c.6268_6269del,), nonsense (c.1633C>T, c.1837C>T, c.2005C>T, and c.5888G>A), missense (c.4448G>A, and c.8245G>A), and splice site (c.6751-1G>A). Additionally, splice site variants were detected in the COL17A1 () and LAMB3 genes (c.8305-1G>A and c.1977-1G>A, respectively)” contains many mistakes please correct.

# In line 53, “predicted These variants” correct it as “predicted these variants”.

# In line 53, the variants are either pathogenic or likely pathogenic but not highly pathogenic, so please correct.

# In lines 54, 55, the authors state that Protein deficiency occurs due to frameshift and truncating mutations, and splice site changes impairing RNA processing. In connection with the above sentence please mention, “what are the consequences of missense variants?”

# The pedigrees are unreadable and illegible. In some of the pedigrees the lines (horizontal/vertical) are missing, in such situations it is very difficult to reach a conclusion about. To make it more clear and easier for the readers it is advised to write the gene name and variant in the left upper side of the corresponding pedigree. It will be also very helpful to write genotype below each individual in the pedigree. Please follow this article DOI: 10.1007/s10528-025-11087-2 as a guide.

# Add OMIM IDs with the gene and disease names.

# Write p.Glu1975fs and p.Pro2090fs frameshift variants in their full form. Such as, p.(Glu1975GlyfsTer29) and p.(Pro2090TrpfsTer8). Also please use uniform nomenclature for c.5924_5927delAACG, c.5924_5927del and other variants like c.1633C>T (p. Gln545Ter), c.5888G>A (p. Trp1963X), c.2005C>T (p.Arg669*), c.1837C>T (p.Arg613*) etc., throughout the manuscript.

# In line 140, it will be better to use GRCh37/hg19 in place of only hg19, because it has already been used in line 211.

# In line 147, add the reference for ACMG guidelines (DOI: 10.1038/gim.2015.30).

# The COL17A1 nonsense variant c.1394G>A; p.(Trp465X) is not mentioned in the text on page 6 under the heading “Genetic Analysis”.

# Write the gene names in italics throughout the manuscript, e.g., in lines 247 etc.

# The “in text-citations” format is not correct, e.g., line 70-- (1) (2) (3), line 93-- (12) (13), line 97-- (14) (12). Write in a correct format such as (1-3), (12, 13), (12, 14).

# In lines 150, 151… segregation of the variant with disease in other affected family members. It will be more appropriate to state “segregation of the variant with disease in other family members”, because the variant may segregate randomly in any of the siblings in heterozygous/homozygous form.

# Use proper reference/URL with the softwares or tools used in the text, e.g., SnapGene version 6.0.2, CADD, FATHMM and SpliceAI etc.

# In line 179, replace “variant forms” with “mutant forms”.

# In line 187, the authors have simply mentioned that “sixteen patients from 12 different families were clinically diagnosed in the clinic” please mention the name of the clinic/s in order to increase credibility and trustworthiness of the research work.

# In line 189, please write the families numbers (11 8 and 12) in a sequence such as 8, 11, and 12. Here it is difficult to confirm that which family belongs to Syria and which one is of Yemeni origin?

# In line 191, the authors have mentioned that “Pedigree analysis (Fig 1) suggested an

autosomal recessive mode of inheritance” but family 12 has autosomal dominant inheritance pattern (have not mentioned).

# In line 194, replace (Fig 2) with Figure 2.

# In table 1, correct the font style in column 7.

# In table 2, correct the gene symbol COL17A as COL17A1.

# For each variant classification based on the ACMG/AMP criteria, write the evidences such as very strong (PVS1), Strong (PS1-4), moderate (PM1-6) or supporting (PP1-5) to be pathogenic or likely pathogenic.

# Correct the heading of table 4, these are the “details of mutations identified in EB patients” add at the end “so far”.

# Correct the sentence “The current study sheds light on the genetic makeup of EB in Saudi patients by detecting causative variants in the COL7A1, COL17A1, and LAMB3 genes” here not the genetic make up of EB but the genetic makeup of Saudi EB patients. So, correct the above sentence. Also replace “sheds light on” with “highlights”.

# It will be more suitable to use “variant” in place of “mutation”.

# No description about the families, consanguineous or nonconsanguineous and how many individuals are present each family etc, which is mandatory to give the detailed description of each family to the readers.

# In lines 360-364 “A total of ten mutations were identified in COL7A1, including three missense mutations (c.4448G>A and c.8245G>A), three frameshift mutations (c.5924_5927delAACG, c.6268_6269del, c.1837C>T), two splice site mutations (c.8305-1G>A, c.6751-1G>A), and two truncating mutations (c.1633C>T, c.2005C>T and c.5888G>A)”. This statement is incorrect because:

Missense: 2 --------(c.4448G>A and c.8245G>A)

Frameshift: 2-------(c.5924_5927del, c.6268_6269del)

Nonsense/truncating: 4------- c.1633C>T; p.(Gln545Ter), c.1837C>T; p.(Arg613Ter), c.2005C>T; p.(Arg669Ter), c.5888G>A; p.(Trp1963Ter).

Splice site: 2--------(c.6751-1G>A; p.?, c.8305-1G>A ;p.?), so correct the above statement.

# The discussion is about the comparison of your results as in this case “the patients’ clinical phenotypes, number of patients in the families, their gender, geography and the identified variants in each family” are logically compared with the previous literature. But here in this manuscript this critical point seems to be missing. The authors are advised to compare their results with those present in table 4.

**Do you want your identity to be public for this peer review?** For information about this choice, including consent withdrawal, please see our Privacy Policy

Reviewer #1: **Yes: ** Muhammad Ajmal

Reviewer #2: No

Reviewer #3: **Yes: ** Dr. Sher Alam Khan

---

## [Author Response · Author response to Decision Letter 1]

14 May 2025

To the Editor and Reviewers,

We would like to extend our sincere thanks to the Editor and all three reviewers for their thoughtful and constructive feedback on our manuscript entitled:

"Identifying Novel Genetic Variants in Epidermolysis Bullosa Among Middle Eastern Arab Families: Insights from Whole Exome Sequencing and Computational Analysis."

We greatly appreciate the time and effort you have invested in reviewing our work. Your detailed comments have helped us significantly improve the quality, clarity, and scientific rigor of the manuscript. We have carefully addressed each point raised and revised the manuscript accordingly. All changes made to the manuscript have been highlighted, and we have provided a detailed point-by-point response below.

We are grateful for the opportunity to revise and resubmit our work, and we hope that the revised version meets your expectations.

Editor Comments:

The findings of the manuscript provide valuable insights on molecular basis of Epidermolysis Bullosa (EB) from Middle eastern families. The study reports novel and known pathogenic mutations in three genes (COL7A1, COL17A1, LAMB3) from 12 families affected with EB by employing exome sequencing. The manuscript is well structured technically sound. However the manuscript requires minor reversions for improvements of quality and better comprehension before acceptance for publication. The following points needs to be considered:

1. As the study reports families from Saudi Arabia, Yemen and Syria, so the title referring ‘ Saudi Arabian families’ is misleading. The use of term ‘Middle eastern’ would have been more appropriate.

Response: Thank you for your valuable comment. Based on the reviewer's comment, we amended the title and text to include Middle Eastern Arab families.

2. The quality of pedigrees in figure 1 needs considerable improvements for clarity.

Response: Thank you for your valuable comment. Figure 1 has been revised to enhance clarity and readability, ensuring that the structure and genotype information are presented more clearly to the readers.

3. A flow chart showing exome sequencing pipeline will be useful.

Response: Thank you for the suggestion. We have now included a flowchart showing the exome sequencing pipeline in the revised manuscript (see Figure 1).

4. More Information on inclusion and diagnostic criteria, details on clinical phenotypes should be added.

Response: Thank you for your comment. We have now included detailed information on the inclusion and diagnostic criteria and clinical phenotype descriptions in the Methods section of the revised manuscript.

5. Some gene mutation nomenclature should be checked, especially the variant described at the protein level. Please follow the guidelines set by the Human Genome Variation Society (HGVS).

Response: Thank you for your valuable observation regarding the gene mutation nomenclature. In response, we have carefully reviewed all listed variants, particularly those described at the protein level, to ensure full compliance with the Human Genome Variation Society (HGVS) guidelines.

The following variants have been checked and formatted according to HGVS recommendations:

- COL7A1, c.5924_5927del (p.Glu1975Glyfs*29)

- COL7A1, c.5888G>A (p.Trp1963*)

- COL7A1, c.1633C>T (p.Gln545*)

- COL17A1, c.1394G>A (p.Trp465*)

- COL7A1, c.6268_6269del (p.Pro2090Trpfs*8)

- COL7A1, c.2005C>T (p.Arg669*)

- COL7A1, c.1837C>T (p.Arg613*)

All mutations now follow the correct HGVS format:

- Protein-level descriptions begin with "p." (without a space),

- Amino acid changes use the three-letter code,

- Frameshift mutations specify the new amino acid (if known), followed by "fs" and the predicted stop codon position (e.g., fs*8),

- Stop codons are represented by an asterisk * as per HGVS standards.

We appreciate your input and have ensured the nomenclature is accurate and standardized throughout.

6. Functional consequences of splice site and missense variant should be discussed in discussion.

Response: Thank you for the comment. We have now included a brief discussion of the functional consequences of the splice site and missense variants in the revised Discussion section.

7. There are grammatical and typological errors in different sections of the manuscript which needs to be corrected

Response: Thank you for your valuable comment. All grammatical and typographical errors have been corrected throughout the manuscript.

Journal Requirements:

https://journals.plos.org/plosone/s/file?id=wjVg/PLOSOne_formatting_sample_main_body. pdf and https://journals.plos.org/plosone/s/file?id=ba62/PLOSOne_formatting_sample_title_authors_affiliations.pdf

Response: Thank you for your comment. We have revised the manuscript formatting and file naming to fully comply with PLOS ONE’s style requirements.

-The name of the colleague or the details of the professional service that edited your manuscript

-A copy of your manuscript showing your changes by either highlighting them or using track changes (uploaded as a *supporting information* file)

-A clean copy of the edited manuscript (uploaded as the new *manuscript* file)

Response: Thank you for your valuable comment. We have thoroughly revised the manuscript for language, spelling, and grammar. It was also proofread by English language professors from the Department of English, Faculty of Arts and Humanities, King Abdulaziz University, Saudi Arabia.

Once you have amended this/these statement(s) in the Methods section of the manuscript, please add the same text to the “Ethics Statement” field of the submission form (via “Edit Submission”). For additional information about PLOS ONE ethical requirements for human subjects research, please refer to http://journals.plos.org/plosone/s/submission-guidelines#loc-human-subjects-research.

Response: Thank you for your valuable comment. Written informed consent was obtained from all participants, and this has been stated in the Methods section.

Response: Thank you for your valuable comment. This publication is based on work supported by Khalifa University (KU) and King Abdulaziz University (KAU) Joint Research Program, Award No. KAUKUJRP-1M-2021. The authors would also like to thank the Saudi Society for Laboratory Medicine for their scientific support.

- This publication is based on work supported by The Saudi Society for Laboratory Medicine for funding the research.

Response: Thank you for your valuable comment. The funding statement has been removed from the Acknowledgments section as follows:

“The authors would also like to thank the Saudi Society for Laboratory Medicine for their scientific support. There was no additional external funding received for this study.

6. We note that there is identifying data in the Supporting Information file “EB Supplementary Tables”. Due to the inclusion of these potentially identifying data, we have removed this file from your file inventory. Prior to sharing human research participant data, authors should consult with an ethics committee to ensure data are shared in accordance with participant consent and all applicable local laws.

-Location data

Response:

Thank you for your guidance. As this is a genetic study, only non-identifiable information such as age (in whole years) and gender was included, as these variables are essential for assessing disease presentation and risk profiles. No personally identifiable information (e.g., names, addresses, ID numbers) was shared. The data were approved for use and sharing in this format by our institutional ethics committee.

7. PLOS requires an ORCID iD for the corresponding author in Editorial Manager on papers submitted after December 6th, 2016. Please ensure that you have an ORCID iD and that it is validated in Editorial Manager. To do this, go to ‘Update my Information’ (in the upper left-hand corner of the main menu), and click on the Fetch/Validate link next to the ORCID field. This will take you to the ORCID site and allow you to create a new iD or authenticate a pre-existing iD in Editorial Manager.

Response: Thank you for your valuable comment. Yes, the corresponding author and co-authors have validated ORCID iDs as listed below:

• Nancy Shehata: https://orcid.org/0009-0006-0237-3478

• Noor Ahmad: https://orcid.org/0000-0002-7133-656X

• Hadiah Bassam Al Mahdi: https://orcid.org/0000-0003-3483-9499

• Shmoukh Abdullah Alghuraibi: https://orcid.org/0009-0005-5552-851X

8. Your ethics statement should only appear in the Methods section of your manuscript. If your ethics statement is written in any section besides the Methods, please delete it from any other section.

Response: We appreciate the reviewer’s observation. In compliance with the journal’s guidelines, the ethics statement has been retained exclusively in the Methods section and removed from all other parts of the manuscript.

9. Please upload a new copy of Figure 1 as the detail is not clear. Please follow the link for more information: https://blogs.plos.org/plos/2019/06/looking-good-tips-for-creating-your-plos-figures-graphics/" https://blogs.plos.org/plos/2019/06/looking-good-tips-for-creating-your-plos-figures-graphics/

Response: Thank you for your valuable comment. Figure 1 has been revised to enhance clarity and readability, ensuring that the structure and genotype information are presented more clearly to the readers. Please note that with the addition of the figure illustrating the WES workflow, the order of the figures has been adjusted accordingly.

Response: Thank you for your comment. As requested, we have reviewed and updated the reference list to ensure it is complete and accurate. All newly added references have been included in the revised manuscript and this rebuttal letter. We also confirm that no retracted articles are cited.

Reviewers:

Reviewer #1: This manuscript will make a nice contribution to the field of human genetics. But needs some improvements and minor revisions before formal acceptance for publication in PlosOne. Authors should consider following points during minor revisions:

1. Authors must provide the transcript IDs of different genes in which mutations have been identified.

Response: Thank you for your insightful comment. We have now included the transcript IDs (RefSeq accession numbers) for all genes in which mutations were identified, ensuring that each variant is accurately referenced according to the correct transcript. This information has been added to the relevant sections of the manuscript and tables, as applicable.

2. In the abstract there are empty parentheses "()".

Response: Thank you for your valuable comment. The empty parentheses in the abstract have been addressed by adding the appropriate variant information to ensure completeness and clarity.

3. Quality of all the figures needs improvement.

Response: Thank you for the valuable suggestion. We have revised and enhanced the quality of all figures to improve clarity and readability. Additionally, we have included a new flowchart illustrating the exome sequencing pipeline in the revised manuscript (see Figure 1).

4. A figure detailing the flow chart of exome sequencing should be made for clear understanding.

Response: Thank you for the suggestion. We have now included a flowchart showing the exome

---

## [Editor Report · Decision Letter 1]

30 Jun 2025

Identifying Novel Genetic Variants in Epidermolysis Bullosa Among Middle Eastern Arab Families: Insights from Whole Exome Sequencing and Computational Analysis

PONE-D-24-59641R1

Dear Dr. Shehata,

We’re pleased to inform you that your manuscript has been judged scientifically suitable for publication and will be formally accepted for publication once it meets all outstanding technical requirements.

Kind regards,

Rabia Habib, Ph.D

Academic Editor

PLOS ONE
---

## [Editor Report · Acceptance letter]

PONE-D-24-59641R1

PLOS ONE

Dear Dr. Shehata,

I'm pleased to inform you that your manuscript has been deemed suitable for publication in PLOS ONE. Congratulations! Your manuscript is now being handed over to our production team.

Kind regards,

on behalf of

Dr. Rabia Habib

Academic Editor

PLOS ONE